# Review on Wearable Technology Sensors Used in Consumer Sport Applications

**DOI:** 10.3390/s19091983

**Published:** 2019-04-28

**Authors:** Gobinath Aroganam, Nadarajah Manivannan, David Harrison

**Affiliations:** Design Department, Brunel University, London UB8 3PH, UK; 1203998@brunel.ac.uk (G.A.); Nadarajah.Manivannan@brunel.ac.uk (N.M.)

**Keywords:** wearable technology, sensors, data, smart, sport

## Abstract

This review paper discusses the trends and projections for wearable technology in the consumer sports sector (excluding professional sport). Analyzing the role of wearable technology for different users and why there is such a need for these devices in everyday lives. It shows how different sensors are influential in delivering a variety of readings that are useful in many ways regarding sport attributes. Wearables are increasing in function, and through integrating technology, users are gathering more data about themselves. The amount of wearable technology available is broad, each having its own role to play in different industries. Inertial measuring unit (IMU) and Global Positioning System (GPS) sensors are predominantly present in sport wearables but can be programmed for different needs. In this review, the differences are displayed to show which sensors are compatible and which ones can evolve sensor technology for sport applications.

## 1. Introduction

### 1.1. Overview of Technology

Technology has allowed greater user-centered design solutions for various industries. There is a growing trend of increasingly quantifying achievements being used by consumers daily, whether it is fitness, health, or work related. Wearables that possess sensors to monitor how the body is maneuvering gives the user greater understanding of themselves [1]. The reason behind the measuring of data can be related to one’s desire to quantify their ability in an area they want to gather more information (tracking) [2]. Advances in sensors allow deeper measuring capability. Users learn more about themselves, thus changes to their lifestyle can be made under their control. Applying sensors to equipment is another way to show progress in sensor technology [3].

The uses for wearables differ. Some need it, some want it. Examples of when it is needed are mostly common in work spaces. An individual will benefit from wearable intelligence if the body worn item can sense relevant data to educate them on a procedure or identify safety concerns [4]. Examples of when it is wanted is when an individual sees an opportunity for personal benefit, gaining valuable data from the sensing’s function (e.g., Fitbit, or tracking how many calories the user burns) [5]. The trend in wearables are crossing that fine line of being wanted to being needed. This is due to most applications becoming more user-oriented, and the data that is being produced to improve oneself. This can start from an interest but adapts to necessity. The adaptation process is based on perceived usefulness against actual ergonomics in wearables (including external factors) [6]. Presently, there is this self-obsessed era where success is being graded on quantity data (heaviest bicep curl rep, number of followers, amount of views, etc.). Factors such as social media are allowing wearable manufactures to thrive in this era [3]. Human centered advancements are making accessibility easier (e.g., voice recognition commands), transforming this way of living to a norm. This only causes an increase in people’s interest to be involved with this sort of application in their daily lives, (e.g., controlling home appliances from a smart watch) [7,8]. 

When a new technology has been trialed and tested, just as it is marketable, the best suited industry will always have it first. Only after this can the wearable become available for other mainstream markets. An example of this is Smart-body worn trackers, which the military benefited from first before they became useful for different purposes [1]. Instances where the same wearables are used in multiple industries means the sensors that are involved produce data that can be processed for different uses, hence different users [9]. Its influence is increasing due to needs being solved constantly by wearable technology. The military and space industries are continuously big influencers in this market [8]. The perception of wearable technology commenced from a computer that’s worn, and is part of the user which is fully controllable, but can work without any thought or effort [10]. This form can somewhat be seen in present day wearables, as they are considered Smart due to operating themselves with minimal human input in controls but also due to giving the user the freedom to take actions from the data that are presented from the wearables.

Fitness possesses a big market share of consumer wearables [11]. This sector is where the wearables are thriving due to people wanting to be at peak fitness levels for health, sport, and aesthetics reasons. A well-known wearable, Fitbit, is a wrist worn device that allows monitoring of the body during exercises and sleep [5]. This wearable can be viewed as a lifestyle device where a user’s priority may perceive them to view fitness as a lifestyle rather than as an activity. Figure 1 elaborates this to show an example of what a current consumer fitness wearable’s process relative to the user would look like. This example explores the multiple technological factors to give users the intended feedback.

### 1.2. Market

The market gives an idea of what the forecast is for wearable tech in all industries. It can show signs of what trends will arise soon. The big two industries are consumer and defense, hence larger investments are required for them [12]. Figure 2 shows how big these two are in relation to others. IDTech mentioned different waves of wearable technology advancement [13]. The first stage is established wearable sensors such as hearing aids, headset, Global Positioning Systems (GPS), cameras, thermistors, etc. These are wearables that have long been used in different industries and have been able to evolve. The second phase is made-wearables. These are micro-electro-mechanical sensors (MEMS) that have been developed for wearables as technology has advanced and are influenced by the impact of smart phones being used daily, such as GPS or inertial measuring unit (composed of accelerometer, gyroscope, and magnetometer) sensors. This has helped fitness tracking wearables revolve around these sensors exclusively. The third phase is the future phase; these are made for wearables, which use advances in research to make better use out of sensors, thus improvements are made in flexibility, motion, and smart textiles [14]. This is where future investments will be heavy, and even though they may not be commercial yet, it is projected to “experience huge growth”. 

Market research shows how fitness and lifestyle products will grow the most for this sector, but wearable technology advancing in multiple industries should not be neglected, as research can be used as crossovers [8]. Gaming is becoming a reputable industry for wearable technology. The emergence of virtual and augmented reality, which is heavily involved in this sector, is playing a part in increasing how a user can immerse themselves with the product and experience. Wearable technology plays a part in this, as it allows senses (via haptics) to be felt, giving a more authentic feel to the experience [15]. These types of innovations are constantly refined aside from the hardware, such as applying wearable sensors on the user themselves. The product’s lifecycle may not involve continuous long hours of usage such as those in the medical or the fitness sectors. Existing technology (same sensors) can still be used in this sector to give different types of users their desired feedback. Game designers will have to identify which senses are authentic for their specific games. 

A simple example of this format is the Nintendo Switch Labo edition. Their Joy cons possess default sensors in accelerometer and gyroscope, which are normally found in consumer wearables. Joy cons and cardboard combine to build their own artificial wearables, such as a robot, to apply it to a gaming function [16]. This is Nintendo’s way of breaking barriers for future generations (who are more involved with technology at younger ages) in immersive experiences with simplified custom wearable tech. Some parents may view this as an ethical solution, as they prefer their children to play with these rather than virtual reality headsets, a popular gaming immersive experience tool. They will also think about social considerations depending on how they can use technology to safely interact with their children [17]. The form of adaptation starts from a wearable that is useful (desired) for an individual, which then moves to wearables for social impact. This is where gamers can interact with their family or friends, and the impact it has on communication with them using a different medium. This will then lead to the perception of the Labo edition being a wearable for public interest, as the potential stakeholders are both the young users and their parents. The process, is the concept of sustainable wearables known to enhance the quality of human life. This structure can be used for any wearable manufacturers that want their designs to become marketable [18]. 

In ergonomic and anthropometric terms, positioning of the wearables is crucial to success. It is fundamental for consumers before purchasing that they know their mobility limitations whilst wearing the device [6]. Figure 3 shows which body part is most popular for wearables, and it supports the idea that wrist wear is very popular amongst wearable designers. This will also influence their adaptation for future wearable technology.

Application of wearable technology in sport comes in different forms. The popular ones are body worn devices (wrist), but there are other types, such as sensor embedded equipment and smart textiles [9]. The goal of these different approaches to wearable technology for sport is to enable the users to have the best possible way of monitoring their performances without hindering any movement. This is also dependent on where the sensor is placed within the equipment, regardless of whether it is an accessory or not [14,20]. In some sport, the equipment having sensors embedded into them would be a better solution than having a wrist worn sensor, such as American football or Ice hockey, where sensors are placed on the pads. Placing them on the shoulder or the helmet can provide more meaningful data due to these sports’ nature of involving those body parts. Location of sensor placement will always be a factor in design as the intended function is prioritized [21]. There are instances where external factors such as the sport’s ball, affect biomechanical factors. Football boots with sensors on the outer sole have been known to produce data that show ball maneuvering characteristics. Sensors can also work well by placing them inside the shoe (insoles) to monitor how the feet react. Relating it to precise biomechanical movements gives the user maximum understanding of their motions. The opportunity to self-learn different ways to improve is a big function in human-centered design [22]. More studies are needed in flexible sensors for consumer wearables to understand how reliable they can be. This is because, for the mass audience, inclusive design principles require extensive human-centered research. Flexible sensors give the opportunity for many wearable design innovations, but the accuracy of processed data can only be judged if they are meaningful, based on consumer feedback. Flexible sensors can be practical if the design of the wearable is ergonomic enough for the user to easily adapt to the change. The advantage of allowing more design ideas to be explored should not compromise the processing of meaningful data. This could depend on other electronic parts, such as the microcontroller, thus testing is the only way to truly know the extent of how impactful flexible sensors can be. 

With material advancements, flexible sensors, conductive threads, and smart material being embedded into clothing, the equipment itself can be a sensor [23,24]. This is a way of including electronics without accessory on clothes, thus the design itself is made exclusively fit for monitoring. There can be an argument as to which industry may benefit from this, however, if the sensors are functioning precisely to their needs, they can be used for many purposes [25]. Giving a greater level of freedom to where key components can go means greater design specification refinement but without compatibility concerns (easier testing phase). This may be useful for training or lifestyle applications to truly test different consumer types for the same wearable, deepening the understanding for engineers and designers for future considerations.

## 2. Sensors

### 2.1. Microcontroller

The key component in allowing wearable technology to function is a microcontroller. This is typically viewed as a mini computer (system on a chip). It enables the Internet of Things (IoT) to be present in this application [20,26]. Importantly, it reduces multiple electronic components that are tasked with performing various functions on a single chip [27]. It is used desirably in wearable technology because of its simplicity to program, reprogram, cost, size, compatibility with other sensors, and the ability to control complex outputs, such as graphical displays [28]. The versatility allows designers to optimize the microcontroller to meet their user needs. 

The microcontroller comprises certain parts. The processor reads the input data and processes the information, which is programmed to make decisions and write output data. The oscillator is a timer clock that syncs all the necessary data. Memory has three parts—random access memory (RAM), which loses data upon power loss, read only memory (ROM), which does not lose data upon power loss, and flash, where the data are programmed for storage. The ports make the final piece of a microcontroller. These are the connections to and from the microcontroller (input/output). It reads the input data (polling) and writes the output data. There is an analog to digital convertor (ADC), which allows analogue input to be processed into digital readings [26,27]. This format of system on a chip is used to identify which architecture the designer would use regarding how the electrode-based sensors require signal conditioning (e.g., analog front end) [23]. 

The microcontroller structure varies for different types. The von Neumann architecture has three buses—the data bus (carries the data between in/out), the address bus (memory location), and the control bus (determines if the data are read or written, others handle interruptions, selects memory or ports). The Harvard Architecture is a faster structure and has an additional fourth bus, the instruction bus (data and instructions are processed at the same time). The default operation is common. Voltage supplied gives all the components power, control logic would take priority of all components, and a timer stabilizes to allow pulse sequences. The program counter is initially set at zero, where the address instruction is then sent. The decoder would read, and then the program counter would go to one. This process is repeated rapidly [26,27]. 

Design considerations that must be prioritized in wearable applications would involve the type of programming. It is important to understand the process of the chosen microcontroller and the process of choosing the features that show input/output conditions. The program language is also important. Visual basic is a common language to program microcontrollers, however, wearable tech engineers are known to use C, C++, Arduino, Java, Swift (Apple IOS), and Python, all dependent on the wearable’s function [29,30]. This makes it easier to set commands to the components, and the function of the microcontroller can be altered rapidly. Reliability depends on the accuracy of the monitored data. These can be the microcontroller itself or the electronic components used alongside them. Different activities can alter the sensor’s accuracy. This could depend on how specific sensing elements work [31]. The sustainability is crucial, because it would determine the lifecycle of the wearable [18]. When there are advances in electronics (e.g., batteries), it influences how new products vary. Material advancements are equally influential in design of the overall system. For the sports industry, wearable technology will always have extensive research in microcontroller, power management of integrated circuits, and how sensor signal conditioning occurs. When these three are defined, the compatibility of sensory elements can be known for a required application. 

### 2.2. Accelerometer

Accelerometers are a common sensor found in wearables. Their sensing capabilities range from different types of accelerations (linear and gravity) [32]. Their measuring capabilities allow monitored data to be programmed for different uses. An example is when a user runs, it can output top speed as well as acceleration. Accelerometers also monitor sleep patterns, which can be linked to seizures [33]. These two examples show that industries such as sport and medical benefit from an accelerometer-based wearable due to its capability in producing a diverse range of meaningful data. It is important that the programmed algorithms must not be affected, as this can corrupt the monitored data during processing phase. This will make the monitored raw data meaningless. [1,12,32,33,34,35,36]. 

Accelerometers can be defined by their limitations; this is normally their maximum capacity of measuring acceleration [36], typically defined as a sensor that can turn kinetic movement into digital measurement. It does this by measuring the accelerative forces. A piezoelectric effect (piezoresistive for direct current) can enable this, as the microscopic crystal structures become stressed due to forces that show a difference in voltage, due to the crystals [37,38]. The other method is via a capacitance difference (capacitive for DC) between two microstructures. This is the same for both classes of accelerometers (analogue and digital outputs). More sensing type variations exist in strain gauges, servo, and vibrating elements [38]. The amount of potential sport attributes that can be sensed via these methods allows the device to know how the user moves, both in accelerations and orientation. The location of the sensors allows flexibility in position, making the accelerometer a very multi-functional sensor [39]. 

### 2.3. Gyroscopes

Gyroscopes are another common sensor found in wearables. They differ from accelerometers in that they measure angular accelerations exclusively. Some applications will prefer to use the accelerometer to determine rotational acceleration, whereas some would want to combine both in order to filter errors. This is to increase the accuracy of the monitored data. There are different types, such as gas bearing, optical, and mechanical [40]. Optical is the one that works differently (non-angular momentum), where the two fiber optic coils are spun in alternating directions, thus they travel different distances, which is monitored (Sagnac effect) [41]. 

Gyroscopes essentially detect angular velocity on their disk. Vibration gyroscope sensors sense angular velocity due to the Coriolis force. These are natural forces that occur due to Earth’s rotation, which acts on a vibrating element [36]. The motion produces a potential difference that gets converted into an electrical signal. This is used for measuring orientation and projection, making it beneficial for stats that involve angles or specific positioning [40]. The method of tracking also means that the location of the sensor can be influential in data collection [39]. If there is a need to integrate the sensors with equipment, another consideration to factor will be selecting compatible electronics and locations around those sensors [42].

### 2.4. Magnetometers

Magnetometers can typically be combined with accelerometers and gyroscopes to form the inertial measuring unit (IMU). Each of these sensors can possess three axes each, depending on the type. It is very similar to what a compass does, and it helps with coordination. Whilst it is normally used with the other two sensors, it complements them by filtering the orientation of the movements [43]. 

Magnetometers measure magnetic forces in relation to Earth’s magnetic field. It does this via the principles of the Hall effect, where, if a current carrying conductor is placed in a magnetic field, then a voltage is generated across the conductor perpendicular to the current and the magnetic field [36]. The electrodes inside the conductor get disrupted (change in density) by the interception of the magnetic field, which results in the voltage reading. If the forces applied changes, then the voltage reading changes proportionately, giving the value and the direction of the magnetic field. This is then given out as an electrical quantity, which gives the orientation due to the vector calculations. Detecting different movements of the same body part gives an extra scale to consider as part of the IMU [9]. 

### 2.5. Global Positioning System (GPS)

GPS is a very common sensor found in multiple appliances (smart phones). It is used for navigation, as it informs users about their location. Data are sent to a satellite where the precise location and time are measured. This works as a transmitter and a receiver, where the information is fed back into the sensor to inform the location [36]. It is used in wearables to measure key data, such as distance, which can be viewed in different ways for different applications. Designers have concerns about the power consumption of these sensors. GPS is useful in team sports, as it eliminates issues that arise with time motion analysis, and coaches can navigate positional team play [44]. This is very important for coaches who prioritize multiple things and may not always provide individually based focus. 

### 2.6. Heart Rate Sensors

There are a variety of sensors and techniques that can measure heart rates. One form is using capacitive sensing where the electrode (sensor) and the human skin, can be idealized as two components that make a traditional capacitor [36]. Photoplethysmography is a phenomenon which uses light to measure blood flow, hence linking it to heart beats [45,46,47]. Fitness trackers, like Fitbit use this method via a photodiode. There is a constant green light emitting onto the skin of the user, where the photodiode can measure the light absorption. This data is converted so a pulse measurement can be processed [5]. The more blood that is flowing through the user’s body, meaning higher intensity, the greater the amount of light absorbed by the diodes [46]. There are methods now, using this phenomenon with red light, to determine oxygen levels [47]. Green and red are commonly used for this purpose, where the “wavelengths” of light depends how “strong the penetration through tissues” can be [45]. 

### 2.7. Pedometers

Pedometers are commonly found in lifestyle-based fitness wearables. They are used to count a user’s steps [48]. This can be in the form of walking or running. There are two versions of pedometers—mechanical and electrical. The latter is the most common form presently and relies on MEMS for accuracy but still works on principles based on mechanical pedometers. The mechanism of the pendulum is used to determine steps. A small metallic pendulum is found in pedometers with two ends, one of which has a weight-hammer. Every step a user makes, the hammer swings and touches the other, then comes back to its initial position. The mechanism is connected to an electronic counting circuit via spring [49]. Initially, there is no current, thus every time the hammer connects to the other side, an open circuit becomes closed. This makes the current flow. When it returns to its initial position, it closes again, restarting the pendulum motion. This enables the circuit to know every step cycle. 

Research conducted by Nebraska Wesleyan University on the accuracy of pedometers suggested that the most accurate readings were given by sensors that only tracked steps compared to activity trackers that have multiple sensors to track other attributes. This means that compatibility of a pedometer is not good in terms of this test. The experiment conducted also noted that there is greater accuracy when the pedometer is worn on waist compared to wrist [50]. This shows the importance of wearable positioning regarding the accuracy of sensory measurement. 

### 2.8. Pressure Sensors

Pressure sensors typically work from strain gauges. When forces are applied on sensors, it produces a resistance change in the circuit. Mechanical quantities such as force are experienced in multiple ways for sport and are converted into an electronic measurement dependent on resistance. This form of measuring strain is done by a Wheatstone Bridge formation, which can detect resistance changes in static or dynamic form [51]. The sensing element can occupy one, two or four of the arms in the Wheatstone bridge formation. This number is dependent on the application of the sensor (how many in compression and tension). The mechanism in sensing allows them to be embedded around equipment to monitor external factors, such as ball contact [22]. It can be used for performance or safety gait monitoring applications because the way force is measured on each part of the foot can determine the distribution. Data can be given on how a player can improve their agility (rate of change in directions) or if they are exerting too much force on one foot or the other [52]. 

Pressure sensors can evolve in the form of using graphene based flexible sensors, which measures how graphene conductive network changes, depending on resistance. Having flexible sensors is important for wearable technology, as it allows sensors to be embedded into equipment as well as smart clothing [24,53]. Barometric pressure sensors are also widely used in smart watches and wearables. They are described as measuring atmospheric pressure relative to the environment, to determine altitude [54]. This sensor is very useful when it comes to monitoring the different elevations that a user goes through during their activities. With body movements, factors such as temperature and airflow are in constant transition. This difference is where barometric pressure sensors are considered useful. They measure these differences as they are compared to relative elements [55,56]. An example of this is when a cyclist’s route turns to an inclined hill road. Compared to a flat road, the altitude will be different; hence the atmosphere change is monitored via barometric pressure sensors. Forecasting weather is another function of this sensor, as it can be used in smart watches [57].

## 3. Electronic Applications

### 3.1. Important Factors

Sensors are the core of wearable technology. Without sensors, there is no use for wearables. Consumers are desiring monitoring systems that give out specific data. These data come from sensors and get processed for the intended user. Figure 4 gives a forecast in the importance of accelerometers, gyroscopes, and the impact of inertial sensors (combined), in their share of the market size. It is projected to be “$2.86 Bn by 2025” [12]. IDTech complemented this research claim in stating that the type of sensory components that will lead revenue for wearable tech (forecasted 2022) show the importance of IMU and optical sensors [13]. This gives an idea of which future phase sensors may need more research. MEMS are very suitable for wearables due to their sizes, as designers prioritize being minimal in weight and power consumption (consequently reducing costs whilst increasing ergonomics). Table 1 shows the sensors that are present in wearable technology for different types of industries. This complements the data from Figure 4, where accelerometers and gyroscopes are heavily present in multiple wearables.

All wearable tech listed in Table 1 has Bluetooth. Data from Vandrico show what each industry possesses in wearable tech and its hardware.

Alongside microcontrollers, there are some essentials for wearable technology to work—how data are communicated (wireless data transfer), storage, and battery [54]. Storage in wearables is dependent on the operating system. The nature of feedback mediums such as smart watches, phones, tablets, or personal computers (PCs) depends on its application. With the use of Cloud storage, wearable firms can send the data from users’ wearables onto their servers for it to be processed (an ethical concern) [17]. With consumer wearables, most applications give feedback on handhelds, meaning there can be a possibility of having the data stored on the phone itself (internal memory/memory card). The time taken to sync the data can vary, meaning there needs to be a base where the storage is kept. This is what can differ between storing on the device itself, e.g., flash, or on servers. Wearable technology manufacturers benefit from using a smart phone, as it possesses electronic components that are useful for wearables, such as Wi-Fi [58]. 

Wireless communication is an essential part of wearables. It is regarded as the wireless sensor network and has different topologies (e.g., mesh, star, etc.) [59]. These work with sensor nodes, which have low maintenance, and monitor the environmental conditions to determine how data transfer would occur [39,58]. This component is fundamental for consumer ergonomics. It also allows the data storage to be defined regarding where the communication should transmit and receive. Radio frequency is commonly used for all essential communication methods. Table 2 shows the different wireless technologies for wearables. Actual quantitative specifications vary for different versions of the same wireless tech. 

Different versions are available for the same wireless tech where parameters are different. The cost refers exclusively to the tag and not the processes involved with development [60]. The bit rates are average values, as they fluctuate depending on application. Design consideration for choosing certain wireless communication methods is dependent on application as well, such as the size or how proof it is in certain conditions (waterproof, shatterproof, UV resistant, etc.) [61]. Bluetooth low energy (BLE) is used greatly amongst the consumer sport manufacturers due to designer’s requiring low cost and power consumption with good reliability [62,63]. Being low in energy consumption will help the sustainability of the network [64]. For wearable tech, it can work in a Piconet, where one master device controls multiple slave devices.

Ideally, the range, the bit rate, and the power consumption are supposed to be proportionate. The specific values can only be known via testing. A 6-Mbits/s Wi-Fi data rate may work around 70 m but can also be 54-Mbits/s for 10 m, and these differ for indoor and outdoor conditions [65]. Most smart watches will work with outdoor standard data sets. The downstream and upstream values also affect wearables, when designing [60]. If a PC is considered as a base station, then the wearable cannot be portable [7]. By having a smart phone paired with it, Wi-Fi can enable the phone to send these data to potential servers. The ranges listed above for Wi-Fi have a variety of versions, each fit for its intended purpose. Ideally, a wearable for the sports sector will not have either of these due to costs and maintenance. Cellular is another type of wireless sensor network, that has very high costs, power consumptions, range, bandwidth, and physical size. Cellular is heavily used in industry and is excluded in the review in Table 2 because it is not a feasible option. However, both Wi-Fi and cellular can be found in lifestyle (smart watches) and industry (google smart glasses) wearables.

Instances arise where manufacturers would choose more than one wireless technology, typically in lifestyle applications. This is because of the diverse use, such as smart watches that sends activity data to a base (BLE) as well as making payments (NFC). Lower bandwidth is preferred for wearables, as most do not need to transfer substantial amounts of data. NFC can also be preferred more in medical wearable sensors. The power consumptions are very dependent on the type of wearable and it’s use in industry [52]. ZigBee has been trialed for gait monitoring experiments and rehabilitation. This was to test its potential in the medical industry with the creation of its mesh networks and in-built security measures, making it a choice as a potential wireless communication method [59,66]. 

The source of power comes from batteries. Evolving battery technologies has helped wearables become the desired consumer items they are today, such as sensors being self-powered [67]. Whilst designing, it is crucial that the battery can be recharged with minimal changes to the wearable itself. This is because it should not need to be unassembled and must have a way to remove the battery without taking the wearable apart too much (modular designs) [68]. The importance in the size of wearables is distinguished by what the designer wants. There is a trade-off between operating time and data quality, which hinders the power source as well as the sensors used. This means that the size of the battery will affect the size of the sensor used. The battery consumption usage can be split into three. The first is the idle state, which can range from 0 to 25% of consumption. Sensing can also range from a similar range. Communication can use up to 50%, however some wearables may send the data whilst sensing, which would make the total (combined) consumption larger [69]. Figure 5 shows the plot taken from Maxim Integrated about battery consumption in wearables. Common types of wearable batteries are alkaline, Nickel metal hybrids, and lithium ion (polymer versions as well), with the latter being the more popular option in wearables. With flexible thin film, energy harvesting is possible due to its high energy density, which can be perceived as another important benefit of Li-polymer batteries (pouch cell) for consumer wearable devices [70]. 

Table 3 illustrates an example of how Fitbit and Viper PODS differ in use. Viper PODS are used for certain periods of the day, whereas Fitbit is used throughout the day. This means that the power consumption states for both these wearables differ. It is important that the designer defines how much power consumption rate the device’s battery will consume. Fitbit monitoring too much can confuse the user if it starts producing data that they do not understand the benefits of. Viper PODS only work during certain hours, which may limit their capacity to judge certain parameters, such as recovery period. When designing wearable tech for the sports industry, it is important to know what the training regime can be for different types of users. This can identify which components are needed. Table 3 shows an example of “a day in the life of…”, a term used to detail a system’s stages throughout the day. Figure 6 displays the example set from Table 3 in how the battery consumption rates change during the day for both wearables. The projections are based on theoretical consumption rates (Figure 5) without specific values to show an example of the difference between Fitbit (lifestyle wearable) and Viper POD (sport specific wearable). The actual percentage of battery consumption during the three states (communication, idle, and sensing) may be completely different. This example is just used to show the difference in the function of two wearables regarding battery used.

There has been research into battery-less wearables [70]. This is for weight reduction (as batteries tend to be heavy) and for better energy conservation (sustainable). A form of harvesting energy, such as using potentially lost energy, can be useful. As with the piezoelectric effect or solar, which use photovoltaic cells (convert photons into energy), there are potential advances in this sector that follow energy harvesting methods [71]. Using kinetic energy (body movement) to charge the wearable is useful, if not obvious, due to the nature of wearable technology applications. These are still hindered by material advancements and how they can be integrated into wearable tech due to size. University of Southampton used piezoelectric energy harvesting methods on the insole of shoes. Charging occurs via kinetic movements. The sustainability of the piezo elements is a liability due to the ease of damage. Solar energies can be used for health purposes as well, such as sunburn time. With the advancement of wireless charging, this can also be used for design considerations where the components can be imbedded into equipment (easy maintenance). 

Having sensors only send and receive data periodically (or in bulk) can help power consumption. This will mean it has to be stored somewhere [17]. Wearable technology integrated with smart phones has an advantage in that it can use the phone’s storage capability. Data need to be kept somewhere safe. Lifestyle wearables depend on continuous readings, and there may be a need to use every data that the wearable are monitoring [28]. This may be something consumers desire. With firms using Cloud services to store consumer data, which has live encryption monitoring, storage priority may not be a concern [68]. However, when ethical issues arise in data privacy, this may make consumers uncomfortable regarding how their personal data are accessible. 

Figure 7 shows an example of how a wearable that has a chip with IMU sensors, can track data and communicate via wireless technology. The option to have it on servers or phones can be dependent on the user, and the feedback is fed into where the designer planned for.

### 3.2. Sensor Calculations

Key formulas linking distance [s], velocity [v], and acceleration [a] via integration/differentiation with respect to time [t]. Integration allows this procedure to work from acceleration to distance in reverse order (bottom-top). Table 4 displays which attributes are monitored by accelerometer and gyroscope, then calculated to produce more data.s[t]: distance is a function of time,v[t] = S’[t]: velocity becomes a function of time (differentiating distance once),a[t] = v’[t]: acceleration becomes a function of velocity (differentiating velocity once),a[t] = v’[t] = s’’[t]: acceleration becomes a function of distance (differentiating distance twice).

The absolute angle can be viewed as the angle of orientation of a body part in reference to a fixed line. This is integrated from the gyroscope measurement (angular velocity). The relative angle is measured by the range of motion that the user experiences. Therefore, it is calculated via the difference [43,72]. The data that are collected from accelerometers or gyroscopes are computed in vector formats [35]. The placement of these sensors is a factor in calculations (mean, variability, standard deviation, matrices, algorithm), which can differ for different sports [31].

### 3.3. Sport Consumer Wearables

Table 5 shows researched examples of sensors in consumer wearable technology for sport. This shows how different wearables rely heavily on accelerometers and gyroscopes for similar purposes, even if they are used differently [54]. 

Table 6 shows how Zepp Play use different combinations of accelerometers and gyroscopes for different applications. This could be due to measuring specific skills that a user could desire. Zepp Play uses BLE for all sensors [73]. It is interesting to see how they position their sensors in such a way that the readings can give allocated data. Biomechanics involved in baseball and golf swings are similar, yet Zepp Play positioned their sensors on the handle of the bat for baseball and on the top of the glove for golf. This could indicate that readings may differ on the position of the sensors for user intended data.

The same sensors are programmed to produce data for different attributes. 3 axes gyroscopes are used in Zepp Play football and golf—two very different sports, yet they measure orientation and are used for two different skill sets [73]. It could be argued that gyroscopes are more influential in golf even though they are known to filter out errors alongside accelerometer readings [48]. Yet, for two similar sports—baseball and golf—in terms of biomechanics of the upper body, there are two different types of gyroscopes used. This indicates that the versions of sensors play a part in data monitoring. Table 7 shows examples of some attributes that IMU sensors track, and which specific sport position they may benefit. This gives a better understanding of where these sensors are being utilized the most for different positions. The sport examples given are a rough indication due to the general skills that could be prioritized. An amateur football defender may prioritize other attributes that typical defenders may not need or want to improve on, thus having a user centered design process helps build trust amongst sport wearable tech consumers. 

### 3.4. Monitored Data

In sport, there is always a subjective opinion needed. This is because the stats that are monitored by sensors are physical [74]. Zepp uses a camera to truly define some of their performance measuring capabilities. Filtering can give the users a representation of their performance [73]. How can sensors measure performance stats? This can be done via data processing, where filters (e.g., Kalman) can convert the physical data in performance terms. Other reliability concerns are whether real time data feedback is as accurate as post game processed data. A study from Victoria University Australia investigated how real time GPS data compared against post game data. There were more errors present in real time tracking, which means that there are still opportunities for electronic improvements for live time accurate monitoring [44]. 

Individual influence in sport varies depending on whether or not it is a team sport. Therefore, there is a complexion in defining some attributes by sensors and whether it can really help team play. Data monitoring shows key skills that can be tracked, but for a sports coach, they will always prioritize the collective team data [74]. Data scientists may always need to be viewing and analyzing the sensory data and linking them to key performance attributes. This may require multiple sensors working together [31]. Performance stats are more technical, and if advancements are made to allow data sync between teammates in training, only then can a collective team progression be made. Even if this is successful, having a subjective opinion is always an important factor (e.g., player may run more distance than usual yet lose a game). Thus, in what context is an individual judged based on team performance, and whether they themselves are improving individually to help the team efficiently or tactically; are questions that can only be subjective [74]. 

Accuracy of wearables and transparency of the data published are fundamental elements. Producing calculations with precisely monitored body movements and quantifying them in a way that consumers understand is a smart procedure, but how well the sensors measure these body movements is another concern entirely. It is still considered that the wearables are not accurate in producing training data [75]. It has been perceived that sensor readings are moderately accurate to actual movements. Accuracy is higher when doing exercises of low to moderate intensities or when doing consistent movements, such as jogging [75,76,77]. This is also the case when the sensors are exclusively measuring one attribute [50]. The accuracy differs more when doing sport related activities where players not only experience high intensity but they are constantly changing agility, which leads to sensors not producing accurate readings [75,76,77]. Readings that users see are not just the quantities that are measured by the sensors but what the program is told to do with these data. The conversions, algorithms, and data process of the monitored quantity are all equally responsible for the accuracy of sensor readings.

Tests have been done on fitness wearables where the accuracy of the data was perceived to be consistent amongst the different types [76,77,78,79]. This validated that the wearables were monitoring accurately but they differed from each other in that they were affected by various activity states [31,76,78]. Examples of variety are when some of the researched wearables produce very accurate readings for step counting, and some give a larger range for heart rates [50,76]. Inaccuracy is judged against one wearable reading. Thus, the difference made by other wearable readings indicates that the sensors used are not consistent enough, giving the range of error. This directly links to what the data conversions are giving, e.g., algorithms [75,79]. When intense activities do occur, the processing of raw data needs to improve in accuracy to give precise meaningful feedback. Because fitness wearables now advertise as tracking many factors, such as energy expenditure, this can complicate the programming side, which results in more inconsistent accurate readings [31,77]. Even if sleep monitoring is accurate, there can be occurrences where the processing of the data is varied due to it tracking so many different activities. Therefore, the wearables must be smart enough to detect the changes of activity states [31,75,79]. Accuracy of sport wearables tend to differ in that they are measured via camera tracking. This method gives a better subjective analysis and makes it easier to quantify biomechanical movement [76].

Design for behavior change is still crucial, even if wearables are considered disruptive tech [77]. There are still people who do not want to wear wearables [80]. Even though technology has become more affordable—smart phones being very dominant in consumer portable belongings—most smart watches are advertised to do what phones do. The question is whether it is a worthy replacement, which is very individual based. Smart phones are a heavy investment. Would it really be possible to replace that with a wrist worn device that functions as a phone or acts as an accessory to it [80]? Due to ergonomic advantages of smart phones, the sustainability of wearables will be compared to them. There can be usability issues that hamper the success of wearables, but wrist worn devices have their own ergonomic advantages [6,79,81]. People may also not want to invest in fitness wearables because smart phones already have applications (apps) that also do what wearables do [80,81,82,83]. The Nike Run app is an example where the smart phone’s sensors (accelerometer, gyroscope, etc.) are slotted into a plastic sleeve, and when the user exercises, it can process the data monitored into physical stats in relation to fitness. Pairing this with Under Armour’s My Fitness Pal app, which takes both food consumption and exercises stats, nullifies the reason to invest in a separate fitness wearable. How users adapt to the experience, forecasts the sustainability of the wearable [77,79].

User experience and interface plays a role in the perception of monitored data [81,82,83]. The user will only judge these results in comparison to their physical activity. If the wearable outputs raw data without making it user friendly, then the perception of accuracy will be questionable. This ergonomic consideration is what accuracy of wearables is judged upon. Where the user sends and receives data is a sustainability factor. Users can share their quantified achievements to relevant people. This can be a motivational reason or a reason to educate themselves more by viewing other’s feedback on progress [83]. Social interactivity increases immersion and learning experiences.

Size and skin contact can be a concern depending on the individual. Aesthetics can be an obstacle to replace traditional watches, which are viewed as jewelry [80]. Older generations may not want them due to their perception on technology advancement [6,79,81]. Fitness enthusiasts may feel they do not need them due to their success from traditional methods. However, there are reasons why consumers would want this technology in their lives in a very self-obsessed era. Thriving on relating success to quantified data is where wearable technology has become disruptive and where trends have spiked since the evolution of social media [3,6,8,79].

### 3.5. Injury Tracking

With increasingly human-centered designs, the use of wearable technology allows industry to work more efficiently, thus funds can be invested effectively. Elderly and disabled will need extensive human factors research, as this is where technology can really help ease a way of living [84]. Remote monitoring can be a solution, as advancements in biosensors have led to this contribution [73]. In the medical field, instead of having a patient book an appointment in advance for a hospital, sensors that allow a recording of the patient’s heart rate (e.g., printed PCB on t-shirt) can allow the doctor to be notified if there are any abnormal activities [21,24]. Figure 8 shows an example block diagram of how wearable technology can be used to monitor health for the elderly and disabled. Known as the wireless body area network, this benefits the user and the doctor, saving time and giving a more transparent form of communication and analysis [59]. Advancements in biosensors have led to this contribution.

The environment is a crucial factor in how smart wearables in the medical sector play a role. In rural areas, the chance of seeing a physician is less than that in an urban area. To have good access to health care, one must travel further if they reside in rural areas. Having a monitoring system that can simplify elements relating to a user’s health condition can benefit them psychologically as well as physically [24]. There is a greater transparency between the condition, the user, and the doctor. However, the user may not prefer this way, as they may not be accustomed to such maintenance of technology and might prefer regular reassurance from an actual doctor [4]. 

Wearable technology in sport is not just about tracking performance. Health monitoring systems that are applied in the medical industry use the same sensors that can be used in sport, allowing wearable technology research to be very compatible [9]. Wearable technology used to assess spine movement is relatively considered as a medical experiment, but its value in sport is just as important [7]. The same sensors can give both the player and their doctor a greater interaction using technology to monitor live time health status. This also educates the player on where they are making it easier to become injury prone. Harbin University’s research into how a multifunctional single sensor is used for bioengineering applications such as gait monitoring and gestures is useful in sport. This is due to the sensors having multifunctional capability whilst reducing complexity [74]. 

American football is known to have sensors embedded into their helmets, which monitors the status of head injuries (concussion) [11,25]. Due to the sport’s nature (frequent head to head tackles), there is a need to monitor how the forces are being dissipated and dispersed throughout the helmet [84]. This measurement can give an idea of how much energy is being felt on the inside of the dampening material (inner foam pads). Material modifications can have potential biomechanical effects, such as how much shock absorption it allows, and pressure distribution [14]. Smart clothing is also an influence in this sector, however, there can be an argument that the protection or monitoring unit may not be aesthetically pleasing or comfortable, which discourages the user from wearing it [85,86,87].

Wearable tech clothing has been researched in improving a baseball pitcher’s biomechanics. This research benefits the performance of the pitcher whilst trying to reduce the chances of potential injuries. Compression shirts are known to be used to detect arm movement and technique [25]. This same method can be used to track the diverse factors of pitching styles, which can relate to performance and injury factors. Sensors are placed in the lower back and arms with conductive threads to give power. Producing data of how pitching consistencies are performed and how injuries can be prevented helps conditioning of the techniques [86]. From Table 6, Zepp golf and baseball editions allow monitoring of the player’s swing (biomechanical features) to improve their stance. This is not just to optimize attributes related to performance (timing, strength, speed, etc.) but also to educate how their movements should be done to minimize chances of injury. North American baseball also has a Health and Injury Tracking System, that displays injury surveillance without wearable tech, but exclusively with observations [88]. Trends are easily noticed this way by collating data on injuries, sessions, body part, position, history, lost time, recovery time, medical clearance, and diagnosis to generate reports for team physiotherapists and doctors to advise coaching staff in training routines. This is a way of using observational methods for monitoring injury patterns. 

There are two types of injury classifications in sport—accidental and overuse. Accidental injuries are sudden, ones that players may not see coming. Accidental injuries have been linked to being traceable from an observational point of view. When a player makes a mistake, they are more prone to rash decisions to compensate, which means they can cause these accidental injuries. Another possibility could be using trackers that monitor sleep patterns; if they are irregular, then it can be linked to making bad decisions based on mental fatigue [58]. These are examples of predicting accidents due to human errors by a player [89]. When an injury occurs, the treatment process can be compiled with medicine, physiotherapy, and adequate rest. This will give an idea of when to undertake gradual training before returning to fitness [90,91]. 

Overuse injuries can be a result of repetitive actions with or without correct form. It can be dependent on the strains and loads, which are applied to certain parts of the body [92]. Minor overuse injuries can heal on their own or with minimum treatment. Major overuse injuries will need extensive care. The intensity at which a player performs can determine how severe an injury can be. A minor injury also has the possibility to become a major injury if the player has not treated it properly. Overuse injuries can affect the bones, muscles, joints, tendons, or ligaments [93]. 

Overuse injuries can be prevented with correct form of movement whilst training, which enables the body to familiarize itself with the correct motions needed [94]. Warming up is renowned as a traditional form of exercise before intense training begins [92]. This allows the muscles to be flexible, strong, and healthy, giving greater blood supply. In terms of cardiovascular strength, low intensity cardio allows the heart rate to gradually increase, which sets up the body to be in a good position to react to drastic changes, such as sudden increase in intensities. These ranges of motion can be done with adequate loads on the tendons and ligaments. Loosening the muscles is needed to give users enough degree of freedom in their movements, passing the strength to resist potential movements that cause joint pain or muscle damage [95]. 

With advances in the medical as well as the fitness sector in wearable technology, there is a good combination for the sports industry to work with in injury tracking and performance. Figure 9 (data adapted from National Health Service UK) shows sport injuries patients in 2012 who were admitted to the Accident and Emergency Department (A&E). A higher number of attendances in sport injuries were 10–19 year old males. This can be classified as youths. The next highest attendees were 20–29 year olds. This means that younger generations may find it beneficial to use smart wearables to reduce the frequency or severity of injuries. These urgent care scenarios are not defined in accidental or overuse, thus a comparison between the two cannot be made.

A study conducted by the University of Birmingham and Southampton FC researched how workload relates to injury in youth level football players [96]. The research revolved around acute chronic ratio (acute workload ratio divided by the chronic work load ratio). This calculation is mostly used to decide when the player can return to their last best-known fitness level. It is used to predict the recovery time of the player and to avoid risks. 

Acute work load is the force experienced by the player in the most recent week of training. Chronic work load is the average force experienced by the player during a 4-week time period prior to the present week (rolling averages). This method is used for different sports, with the characteristics being forwarded to different situations. Gradual increases in loads and intensities must be taken with precaution to reduce the chance of overuse injury [96]. Mechanical loads are defined by the sensors as the cumulative index of effort based on acceleration [97]. This format is trying to build a resistance to the loadings at an adaptable pace. If the acute loadings spike abnormally, then they are likely to cause injuries. It is important to gather data of players before they train to know their different states and history of injuries [90]. 

If a wearable device has been preprogrammed to measure these loads, then certain conditions need to be applied. This is where human input is very important, e.g., if a player has not been putting in as much effort during the first two weeks for psychological reasons and then manages to increase their efforts, the tracking device could show a chance of injury because there were very minimal readings considering rolling averages. This hinders the accuracy of the wearable in feedback, even when the sensors are working perfectly. The player could have been accelerating at a normal rate during times where the device may not have been worn. The choices that a player makes can make a wearable’s judgment less reliable. It requires precise and sensitive monitoring throughout the day to fully define chronic load injuries. Essentially, the acute to chronic ratio helps player conditioning and prevents injury whilst allowing players to perform efficiently to an extent [96]. 

The acute to chronic workload ratio is regarded as a better predictor of injury than either acute and chronic alone. Hulin et al. conducted a study into rugby league players, finding that higher chronic workload players are more resistant to injury in moderate ratio conditions [98]. They are more prone to injury when the acute to chronic ratio is very high. These can be causes of those sudden spikes in abnormal motions during training. It can be noted that sensors have shown enough characteristics to predict elements that can identify high risk movements [96]. Bath university worked on Rugby Union players and how they have a decreasing rate of thresholds, i.e., the maximum load that a player can exert before injury, as the season progresses due to fatigue [99]. Therefore, conditioning during preseason is necessary, as players can increase their threshold limits. Combining multiple key elements will help physiotherapists know how a player can recover. The important question designers will face is how this information can be relayed back to an amateur athlete who solely relies on wearable technology feedback [97]. 

Table 8 shows some potential examples of which biomechanical factors in sports that lead to injury can be monitored by sensors. Consistency is the key to how a gradual work load increases safely. This method of defining injury via quantifying workloads means that if a player is accelerating at a higher level continuously for over two weeks, the player is regarded as having a greater chance of injury. This higher level can only be determined by how the wearable defines the user’s average acceleration. This reading must be crucial if the wearable can measure the user’s change in average levels. This could then distinguish if the player is slowly improving their work load without risking themselves to injury. Table 8 also restates how important IMU sensors can be. They are heavily used in sports wearables for performance, but this table shows how these sensors potentially show signs of injury monitoring. This is where data processing is a complex feature, as it needs to be able to derive these parameters and distinguish the difference in what it measures. Even when analyzing running, there is a need to know the biomechanical features to determine how to increase performance or reduce injury. This is where the advancement of both accelerometers and gyroscopes together is useful [42]. For some examples, this can be simple, such as a gyroscope being used to determine the angular rotation of the hip, which can immediately show if it is too excessive. Calculated and programmed values can be used in acute chronic ratio based on monitoring rolling averages that lead to a potential injury. 

Presently, it is considered that if the ratio is large (greater than one), it means that the acute workload is greater in the current training week [96]. If a player experiences an overuse injury, would there be an option to implement this data in wearable tech? How features in equipment design can be something desirable for the consumer for tracking such parameters is a question for the near future [100]. There are numerous factors to determine how frequent overuse injuries occur for a player [92]. Sensors can monitor total distance covered, but observations can show external factors when wearable technology is not worn. The longer the player runs, the sorer their muscles will be; due to endurance, there is a higher chance of injury based on observation. This would link to how injury prone the observed player could be. Multiple factors must be combined to determine overuse injury, such as length of high intense loads and the momentum, which is linked to the mass and the speed of the player. These can be used to forecast the length of time a player has been running at higher intensity and the more frequent loads per step they experience, consequently giving a better understanding of where injuries could occur. It should not be mistaken for gradual improvements. GPS trackers on wearable technology are known to monitor load values such as the Viper POD [96,101]. Monitoring how the average peak impact of each step on both feet can show where the user may be more prone to injury (left or right side) [102]. It helps users maintain correct form and improve efficiency. Therefore, the programming of wearable technology (microcontroller data processing) is vital [103,104]. 

Data protection is an ethical concern. There are security measures deployed to prevent consumer data from being accessible by other parties. This is an important parameter that designers and manufacturers of wearables will have to consider if they are to release it to the consumer market. This is important not only to protect the consumer’s identity and data relating to their personal wellbeing but also to protect the programming of the system. If hacks do occur on a manufacturer’s system, then it is very easy to alter the algorithms, which can completely disrupt the data processing elements of the wearable (reading out wrong data to confuse or worry the user). Therefore, balancing the budget is critical in terms of how many layers of cryptography the designer wants. Psychology plays a big role in sports performance, and momentum is perceived as the consistency of good form, but how can these terminologies be monitored and implemented into wearable tech? These terminologies are heavily used in observational stand points, and such progression is still to be made.

## 4. Review

Designers will always be analyzing their best options, which are dependent on a variety of factors. Size of the wearable is important in sports, as it should not inhibit movements of players’ motion. This would also correlate with its weight. This can determine where to position the wearable. The materials will also influence how much these factors would compromise the wearable. Battery life and power consumption can be rectified by defining how the wearable is primarily going to be used. Compatibility and wireless area network technology range shows how the wearable can communicate and work with other devices. The type, amount, accuracy, and location of sensors are the core fundamentals. Without researching this, the tracked data will be meaningless to the consumer. Knowing how to process this data for user benefit will depend on programming language. This is where the user experience, feedback, and interface will be crucial. How can the consumer truly understand the data in order to improve the attributes they want? How does the designer deliver this in a human-centered way? These are constant questions in this field. This is something that the designer can only reiterate via consumer study and psychological feedbacks in user interface. This leads to the method of an operating system being used. Wearable technology can help consumers to immerse themselves in their desired sport and improve themselves at their control. More research questions need answering, such as how to decrease the injury frequency in youth players for different sports. How can wearable technology be used to improve desired attributes for performance through meaningful feedback? 

In conclusion, this review outlines the importance of sensor advancements. Research into the priority of sensors can help us to understand where progression in electronic components can be most effective. Listing out the main overview of the technology and the market trends provides an idea as to what the current and the future state of wearables are. Certain sensors are more important than others, and different industries utilize different applications. Design considerations are influenced by current technological constraints. It is fundamental to outline key elements that revolve around consumer wearables in hardware to understand how data are managed in sport terms for performance and injury monitoring. Software is even more important, as the feedback is where the usefulness of the wearable is truly known. This review paper conveys that more research can be conducted into how the specific type of sensors give different readings and that future studies into user experience could be very helpful.

## Figures and Tables

**Figure 1 sensors-19-01983-f001:**
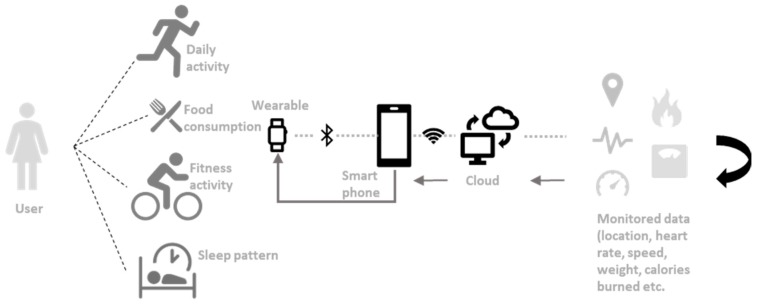
Block diagram example of a Fitness wearable’s process. This shows the example block diagram in how fitness wearable technology can be used for lifestyle applications (weight, calories burned, heart rate, speed, etc.). The users will have their activity monitored via sensors, input some of the data themselves (food eaten), which can then be communicated to a smart phone and to the provider’s cloud service. The data then get processed, thus they become useful for the user to understand. This is fed back into either the paired smart phone or the wearable itself, depending on type of display.

**Figure 2 sensors-19-01983-f002:**
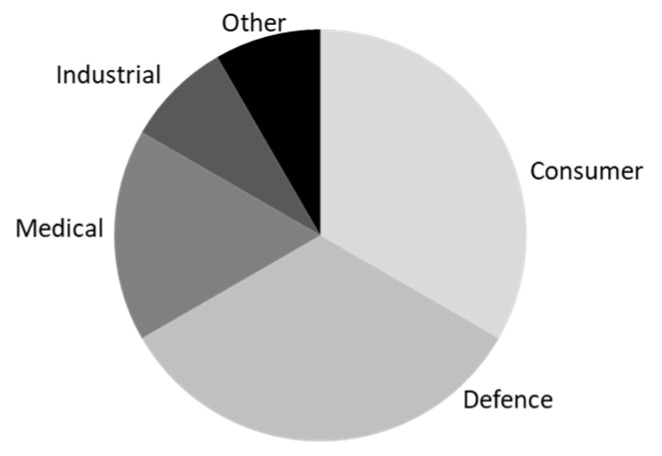
Wearable technology market share; data adapted from Grandview Research [12]. This shows how the global wearable sensors market share is divided.

**Figure 3 sensors-19-01983-f003:**
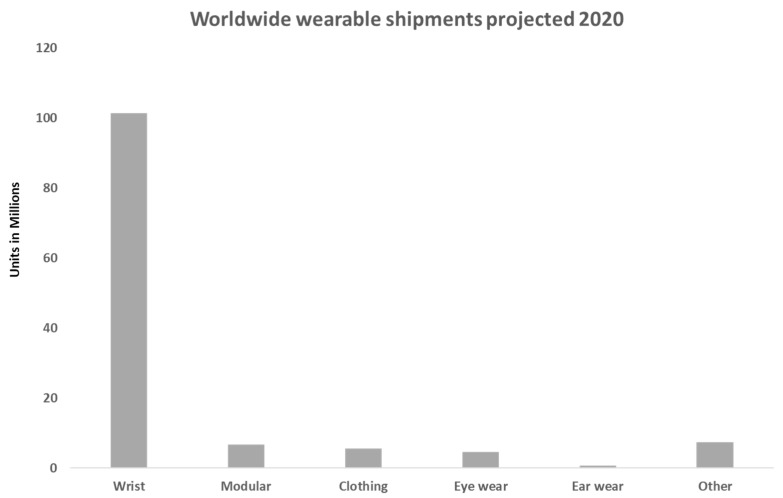
Worldwide wearable shipments projected 2020; data adapted from Statista plot [19]. This figure shows how wrist wear will remain a popular wearable. This could be due to it replacing traditional watches, which consumers have worn for years. If this forecast continues past 2020, then more manufacturers will look to improve on this element alone, but depending on positive consumer feedback, they may even find another body part that can be used for future device placements.

**Figure 4 sensors-19-01983-f004:**
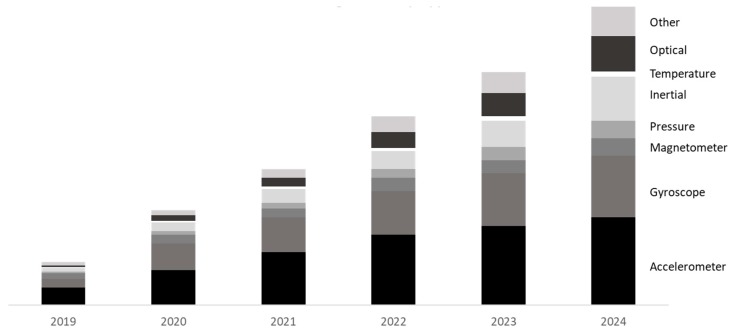
Sensors market share; data adapted from Grandview Research [12].

**Figure 5 sensors-19-01983-f005:**
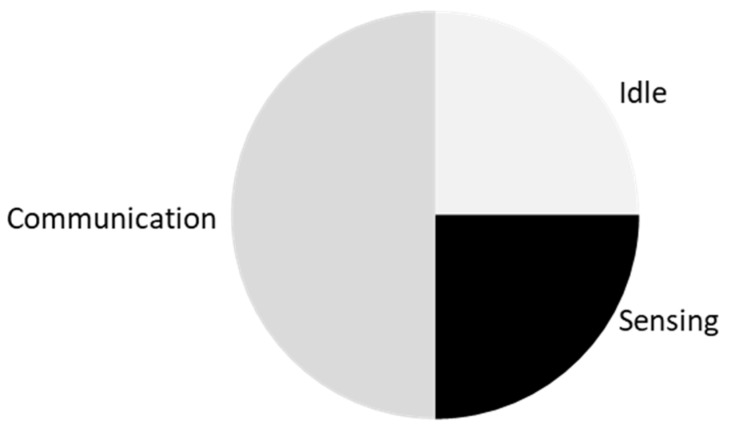
Battery consumption states in wearables (%); data adapted from Maxim Integrated [69].

**Figure 6 sensors-19-01983-f006:**
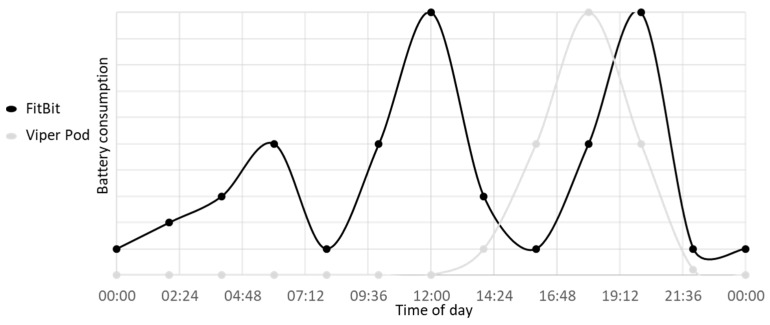
Example of potential battery consumption rate changes during the day for both wearables.

**Figure 7 sensors-19-01983-f007:**
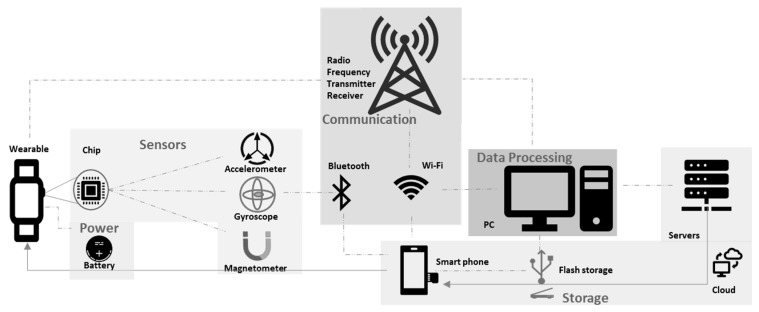
Block diagram example of wearable technology framework.

**Figure 8 sensors-19-01983-f008:**
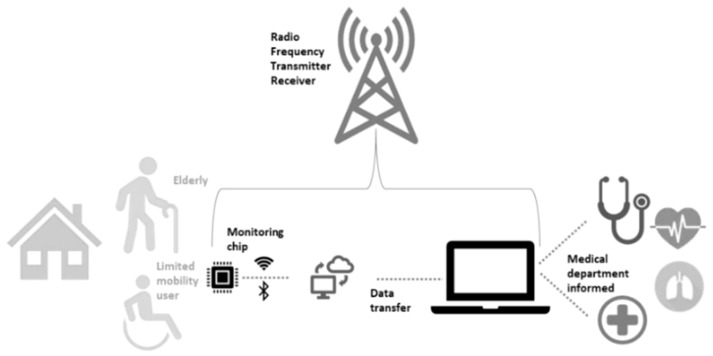
Block diagram example of a medical wearable’s process.

**Figure 9 sensors-19-01983-f009:**
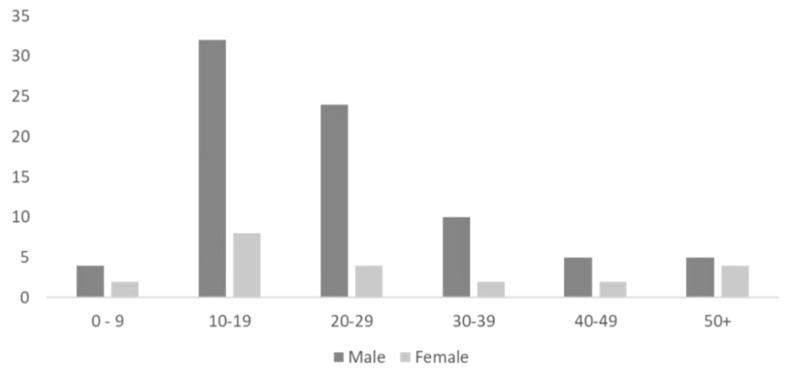
Percentage of A&E attendances for sport injuries by age group; data adapted from Nhs.uk [93].

**Table 1 sensors-19-01983-t001:** Wearable technology in different industries; data taken from Vandrico [54].

Wearable	Accelerometer	Gyroscope	Heart Rate Monitor	GPS	Smart Category	Application	Body Place	Other Sensors
Apple Watch 2	x		x	x	Watch	Lifestyle	Wrist	Speaker
Fitbit	x		x	x	Watch	Fitness	Wrist	Photodiode
Nintendo Joycon	x	x			Controller (modular)	Gaming	Hand *	IR sensor, NFC
PlayStation VR	x	x			Eye wear	Gaming	Head	Microphone, speaker
OM Bra	x		x	x	Clothing	Medical	Upper body	Pedometer
RealWear HMT	x	x		x	Ear wear	Industrial	Head	Microphone, Speaker, camera
HexoSkin	x		x		Clothing	Fitness	Upper body	Pedometer, ECG sensor, Thermometer
Vuzix AR3000	x	x		x	Headwear	Medical	Head	Camera, Magnetometer, microphone
Google Glass	x	x		x	Eye wear	Industrial	Head	Magnetometer, microphone, speaker, light sensors, IR sensor, Camera
Samsung Gear S3	x	x	x	x	Watch	Lifestyle	Wrist	Barometer, Light sensor

* Nintendo Joy con differs on body place, as the Labo edition allows the Joy Cons to be placed on any slot depending on the game [16].

**Table 2 sensors-19-01983-t002:** Wireless technologies for wearables; data taken from their respective websites.

Wireless Technology for Wearables	Cost ($)	Power Consumption	Range (m)	Bandwidth	Bit Rate (Mbit/s)	Physical Size	Wearable Industry
Bluetooth LE	5–35	Low	~100	Low	0.12–2	Small	Sport
Near Field communication	25–100	Low (higher with passive tag)	~0.2	Low	0.4	Small	Medical Lifestyle
Bluetooth classic	5–35	Moderate	~100	High	1–3	Small	Lifestyle
ANT	15–40	Low	~30	Low	0.12–0.6	Small	Sport
ZigBee	4–20	Low	10–100	Low	0.25	Small	Industrial
Wi-Fi	50–120	Very high	10–70	High	2–54	Small	Industrial Lifestyle

**Table 3 sensors-19-01983-t003:** Example in comparing different stages of Fitbit and Viper PODS throughout a day.

Wearable	06:00	09:00	12:00	15:00	18:00	21:00	24:00	06:00
Fitbit	Wake up	Eaten, travelled to work	Eating lunch, walk outside of work	Been at desk for 2 h (idle)	After Work finished, Workout at Gym	Dinner eaten, resting at home	Already in Sleep	Wake up
Viper Pod			Analyzing previous performance	Charging, setting up for today	Training starts	Training finished, analysis feedback		

**Table 4 sensors-19-01983-t004:** How accelerometers and gyroscopes work in computing desired sport physical attributes.

Sensor	Acceleration (m/s^2^)	Velocity (m/s)	Distance (m)	Angular Velocity (rad/s)	Angular Acceleration (rad/s^2^)	Relative Angle (rad)	Absolute Angle (rad)	Force (N)	Moment (Nm)
Accelerometer	Measured	Derived	Derived (2x)	-	-	-	-	Mass derived	
Gyroscope	-	-	-	Measured	Derived	Difference calculated	Integrated	-	Inertia derived

**Table 5 sensors-19-01983-t005:** Sensors found in consumer wearable technology for sport; data taken from Vandrico [54].

Sports Wearable	Accelerometer	Gyroscope	Magnetometer	Heart Rate Monitor	GPS	Position
Fitbit	x		x	x	x	Wrist
Zepp Play	x	x			x	Equipment *
Lumo Run	x	x			x	Lower back
Optimeye	x	x	x			Back (Vest)
PlayerTek	x		x		x	Back (Vest)
Viper POD	x	x		x		Back (Vest)
Adidas MiCoach	x	x				Ball

* Zepp Play uses their sensors for four different sports. They link the sensory findings to technical attributes for specific sports. GPS = Global Positioning System.

**Table 6 sensors-19-01983-t006:** Accelerometer and gyroscope combinations in Zepp wearable for different sport applications; data taken from Zepp [73].

Zepp Play	Accelerometer Type	Gyroscope Type	Position	Sport Specific Attributes Tracking
Football	3 axis accelerometers	3 axis gyroscopes	Calf	Sprints, Distance, Kicks, Top speed, Loads
Baseball	Dual accelerometer	Dual 3 axis Gyroscope	Handle of Bat	Bat speed, Accuracy, projectile, hand speed, attack angle, vertical angle, time to impact
Golf	Dual Accelerometer	3 axis gyroscopes	Top of glove	Club speed, Hand plane, Downswing, Backswing, hip rotation, Tempo ratio
Tennis	Dual accelerometer	Dual 3 axis gyroscope	Handle of racket	Ball speed, ball spin, serve, forehand/backhand, topspin, drive, active time, calories, slice

**Table 7 sensors-19-01983-t007:** Example of attributes that inertial measuring unit (IMU) sensors can track for specific positions in different sports.

Attributes IMU Sensor Measures	Football	Athletics	Baseball	Basketball
Number of Sprints	Strikers, midfielders	Relay	Base runners	All
Vertical acceleration	Forward wings, Full backs	Sprinters, Marathon runners, long jump, high jump	All batters	All
Top speed	Strikers, forward wings, wing backs, defenders	All runners	All batters, short stop, outfielders	Power/small forwards
Distance	Forwards, midfielders, defenders	Marathon, sprinters, relay	Outfielders	All
Intensity Distance	Forwards, defensive midfielders	Sprinters, relay, marathon, heptathlon	Shortstop, outfielders, all batters	All
Vertical Jump	Forwards, defenders, goalkeepers	High jump, long jump, hurdles, heptathlon	Outfielders, shortstop, all basemen	All
Horizontal jump	Goalkeepers	High jump, Long jump, heptathlon	Short stop, all basemen, catcher	Point guard, post
Hand speed	Goalkeepers	Heptathlon, javelin	Power bats, pitchers	Forwards
Hip rotation (kick speed)	Power Kick specialists	Javelin, heptathlon	Pitchers	Post
Trajectory	All (freekick specialists)	heptathlon	All batters, pitchers	Shooting guard
Backswing	Power kick specialists	High jump, long jump, hurdles, heptathlon	All batters	All
Forward swing	Power kick specialists	High jump, long jump, hurdles, heptathlon	All batters	All

**Table 8 sensors-19-01983-t008:** Example of potential incorrect maneuvers that leads to injury which can be monitored via sensors for different sports.

Biomechanical Factors Leading to Injury	Sport	Motion and Possible Injury Example	Sensors
Falls	ALL	Dangerous drop of body weight or collision	IMU
Excessive loads on leg (feet, knees)	Football Basketball American football/Rugby Cycling/Athletics Ice hockey/Figure skating	Dangerous running methods Excessive jumping (vertical) Incorrect agile sprints Unbalance loads on feet Dangerous skating elevations	IMU, pressure
Excessive load on arm (forearm, biceps, hands)	Boxing Baseball Basketball	Incorrect contact elevation Consecutive fast ball pitching Slam Incorrect pitching balance can lead to loads on joints Dunking too hard	IMU, pressure
Stress	ALL	Irregular heart/respiratory rate, blood pressure	Heart rate monitor, IR sensor
Arm speed	Tennis Baseball/Golf	Incorrect rapid dangerous incident angle swings Sudden irregular forward swings	Gyroscope, Accelerometer
Kick speed	Football/American football/Rugby	Improper technique can lead to cramps	Gyroscope, Accelerometer
Angular Collision	American football/Ice hockey/Rugby	Due to tackling nature, concussions occur	Gyroscope, Accelerometer
Excessive rotation arm	Baseball Tennis Boxing	rapid curve ball/slider pitching/Backswing motion (batting) Dangerous serving, back hand strokes. Too fast in combinations can exert force incorrectly on joints	Gyroscope, Accelerometer
Excessive rotation leg	Cycling Football/Rugby/American Football	Incorrect balance whilst pedalling Vigorous kick elevation can cause muscle strains	Gyroscope, Accelerometer
Abnormal body temperature	ALL (water sports included)	Change in body temperature can be due to an accident or environmental conditions	Thermistor GPS

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
