# Peer review of "Review on Wearable Technology Sensors Used in Consumer Sport Applications"

_sensors, 2019, doi:10.3390/s19091983_

Reviewer 1 Report

This paper reviewed a broad range of different wearable sensors based technologies in consumer sport area, and their roles in delivering useful data in different applications. The paper is well organized and the work is comprehensively described. The English is well written in general. But minor improvement on the language with the first one or two sentences of many paragraphs, which are currently too generic and less focused to the topic, will be highly recommended.

Author Response

Thank you for taking your time in reading my review paper. I valued your opinion greatly and have made amendments.

This paper reviewed a broad range of different wearable sensors based technologies in consumer sport area, and their roles in delivering useful data in different applications. The paper is well organized and the work is comprehensively described. The English is well written in general. But minor improvement on the language with the first one or two sentences of many paragraphs, which are currently too generic and less focused to the topic, will be highly recommended.

Response: I made a thorough grammar check and have made sure that the whole text flows smoothly with nice short sentences, and deleted type errors.

Reviewer 2 Report

Overall this review is well written. There are a few points the authors should consider.

1) There are several minor typos/grammatical errors that should be addressed before publication

e.g., "on wearables market",  "diagram in how", "providers 'cloud' service" (provider's), "growearable technologyh", "Football boots having sensors on the outer sole, have been known to produce data that shows ball maneuvering characteristics, which can work well with the sensors placed inside, in how the feet reacts to it and relate to precise biomechanical refinement, giving the user maximum understanding of their movement, hence to how to improve" (somewhat of an awkward sentence).

2) I think the paper would be much improved with more discussion on the accuracy of these commercial sensors. For example, the authors state: "more accurate readings from footsteps" on page 11 but really don't talk about more accurate than what? In fact, a section on accuracy results of commercial sensors would increase the impact of the review paper substantially in this reviewer's opinion.

3) Also, the paper seems somewhat thin. For example, the authors state on page 5 that: "Different industries such as sport and medical can benefit from an accelerometer-based wearable" but cite only one paper. As the authors claim: "Accelerometers are the most common sensor found in wearables", then a greater review of the literature is needed.

Author Response

Thank you for taking your time in thoroughly reading my review paper. I valued your opinion greatly and have made amendments.

1) There are several minor typos/grammatical errors that should be addressed before publication

e.g., "on wearables market",  "diagram in how", "providers 'cloud' service" (provider's), "growearable technologyh", "Football boots having sensors on the outer sole, have been known to produce data that shows ball maneuvering characteristics, which can work well with the sensors placed inside, in how the feet reacts to it and relate to precise biomechanical refinement, giving the user maximum understanding of their movement, hence to how to improve" (somewhat of an awkward sentence).

Response 1.       Typos have been addressed, most of these occurred when formatting onto MPDI template. I apologise for this as I should have proof read thoroughly before submitting. Line 141, I have rephrased the football boots (awkward sentence)....

2) I think the paper would be much improved with more discussion on the accuracy of these commercial sensors. For example, the authors state: "more accurate readings from footsteps" on page 11 but really don't talk about more accurate than what? In fact, a section on accuracy results of commercial sensors would increase the impact of the review paper substantially in this reviewer's opinion.

Response 2.       I have made a section just to speak about accuracy (Page 14 – Line 510) . There were 4 papers that really compared a lot of the “fitness wearables”, and the themes that came with it led onto design for behaviour change, user experience and adoption process. User experience was a bigger part of this, as the papers showed how influential data processing can affect potential readings. These readings are the ones that are judged on accuracy.My own testing on the accuracy of fitness wearables was my 3rd potential review paper. Hence I did not want to touch on this in too much detail.

3) Also, the paper seems somewhat thin. For example, the authors state on page 5 that: "Different industries such as sport and medical can benefit from an accelerometer-based wearable" but cite only one paper. As the authors claim: "Accelerometers are the most common sensor found in wearables", then a greater review of the literature is needed.

Response 3.       For the accelerometer part, I added relevant references and edited the whole section to signify how meaningful the readings are from accelerometer in 2 seperate industries. The rest of the review paper (Figure 4, table 1, table 5, table 6, table 8), showed enough evidence that accelerometers were more common than other sensors, in wearables. 

Reviewer 3 Report

The paper systematically investigated wearable technologies and applications in sport science, however some minor points need to be improved: 

1) Line 29-30, what is meaning of "these are also in ...."? Not clear. 

2) In line 85, "technologyh" seems to be a typo error. 

3) The author should check whether all figures and tables are linked in the text, It seems the sentences citing figure 1, 2, 3 ... are missing. 

4) The author should highlight the challenges of wearable devices in the current stages. i.e. why do some users not like using wearable devices? 

5) In line 138-146, is flexible wearable sensor practical in daily using? is it reliable enough? 

6) In line 181-185, the sentences are not clear, such as "reliability hinders the accuracy...."? Is it related to microcontroller? 

7) The author only talk about pressure sensor in line 265-280, it will be better also discuss about barometric pressure sensor, since it has been already widely used. 

8) In line 376, could the author explain why fibit monitoring too much can be a problem? 

9) In line 470, title "data management" seems to talk about database technology, not related to the content?

10) In line 500, Figure 10 does not exist, should be "8". 

11) Figure 9 is also not cited in the text. 

Author Response

Thank you for taking your time in thoroughly reading my review paper, and referenced it clearly, it helped a lot to find where you wanted me to make amendments. I valued your opinion greatly.

1) Line 29-30, what is meaning of "these are also in ...."? Not clear.

Response 1.       Rephrased

2) In line 85, "technologyh" seems to be a typo error.

Response 2.       I have edited the typos

3) The author should check whether all figures and tables are linked in the text, It seems the sentences citing figure 1, 2, 3 ... are missing.

Response 3.       Figures are cited in text

4) The author should highlight the challenges of wearable devices in the current stages. i.e. why do some users not like using wearable devices?

Response 4.       Line 536; added the section for why consumers may not want to buy wearables, and the sustainability issues that can come with them. Just before this I had to speak about accuracy concerns as this was a good way to leaning towards adoption and design for behaviour change.

5) In line 138-146, is flexible wearable sensor practical in daily using? is it reliable enough?

Response 5.       Line 143-154 speaks about the possible challenge for flexible sensors in consumer wearable tech, and how there are potential advantages too.

6) In line 181-185, the sentences are not clear, such as "reliability hinders the accuracy...."? Is it related to microcontroller?

Response 6.       Line 195-199 is more clearer now

7) The author only talk about pressure sensor in line 265-280, it will be better also discuss about barometric pressure sensor, since it has been already widely used.

Response 7.       Line 296; added section for barometric pressure sensors

8) In line 376, could the author explain why fibit monitoring too much can be a problem?

Response 8.       Line 402; rephrased the part where Fitbit monitoring too much can cause data process confusion to the user itself.

9) In line 470, title "data management" seems to talk about database technology, not related to the content?

Response 9.       Line 495; I replaced the title “Monitored Data”, as this speaks about the overall elements associated with the data that’s being outputted by the wearables

10) In line 500, Figure 10 does not exist, should be "8".

Response 10.   Figure 10/12 – replaced with Figure 8 – Table 8

11) Figure 9 is also not cited in the text.

Response 11.   Line 652- Figure 9 was already citied in Text

Round  2

Reviewer 2 Report

I have no further major comments. Another proof read may be helpful (e.g., "Readings that users' see" rather than "Readings that user sees"); ("aren’t" may be too informal for an academic publication - "are not" may be better instead).